# Probing the E1o-E2o and E1a-E2o Interactions in Binary Subcomplexes of the Human 2-Oxoglutarate Dehydrogenase and 2-Oxoadipate Dehydrogenase Complexes by Chemical Cross-Linking Mass Spectrometry and Molecular Dynamics Simulation

**DOI:** 10.3390/ijms24054555

**Published:** 2023-02-25

**Authors:** Oliver Ozohanics, Xu Zhang, Natalia S. Nemeria, Attila Ambrus, Frank Jordan

**Affiliations:** 1Department of Biochemistry, Institute of Biochemistry and Molecular Biology, Semmelweis University, 1094 Budapest, Hungary; 2Department of Chemistry, Rutgers University, Newark, NJ 07102, USA

**Keywords:** 2-oxoglutarate dehydrogenase, 2-oxoadipate dehydrogenase, dihydrolipoyl succinyltransferase, protein–protein interactions, chemical cross-linking mass spectrometry, molecular dynamics simulations

## Abstract

The human 2-oxoglutarate dehydrogenase complex (hOGDHc) is a key enzyme in the tricarboxylic acid cycle and is one of the main regulators of mitochondrial metabolism through NADH and reactive oxygen species levels. Evidence was obtained for formation of a hybrid complex between the hOGDHc and its homologue the 2-oxoadipate dehydrogenase complex (hOADHc) in the L-lysine metabolic pathway, suggesting a crosstalk between the two distinct pathways. Findings raised fundamental questions about the assembly of hE1a (2-oxoadipate-dependent E1 component) and hE1o (2-oxoglutarate-dependent E1) to the common hE2o core component. Here we report chemical cross-linking mass spectrometry (CL-MS) and molecular dynamics (MD) simulation analyses to understand assembly in binary subcomplexes. The CL-MS studies revealed the most prominent *loci* for hE1o-hE2o and hE1a-hE2o interactions and suggested different binding modes. The MD simulation studies led to the following conclusions: (i) The N-terminal regions in E1s are shielded by, but do not interact directly with hE2o. (ii) The hE2o linker region exhibits the highest number of H-bonds with the N-terminus and α/β1 helix of hE1o, yet with the interdomain linker and α/β1 helix of hE1a. (iii) The C-termini are involved in dynamic interactions in complexes, suggesting the presence of at least two conformations in solution.

## 1. Introduction

The human 2-oxoglutarate (α-ketoglutarate) dehydrogenase complex (hOGDHc) is a key enzyme in the tricarboxylic acid (TCA) cycle, comprising multiple copies (subunits) of three components: the 2-oxoglutarate dehydrogenase (hE1o), dihydrolipoyl succinyltransferase (hE2o), and dihydrolipoamide dehydrogenase (hE3), with an overall molecular mass of ~4 MDa. The hE1o with tightly but not covalently bound thiamin diphosphate (ThDP) decarboxylates 2-oxoglutarate to a reactive intermediate, the enamine or C2α-carbanion, which then reductively succinylates the lipoyllysyl arm of the hE2o; next, the succinyl group is transferred via a transthiolesterification mechanism to CoA producing the succinyl-CoA product; finally, the hE3 component reoxidizes the dihydrolipoyllysyl-hE2 to lipoyllysyl-hE2 producing an equivalent of NADH in the process and setting up the complex for the next turnover (Figure 1). The OGDHc represents one of the main regulators of mitochondrial metabolism through NADH and reactive oxygen species (ROS) levels and affects cell metabolic and signaling pathways through the coupling of 2-oxoglutarate (OG) metabolism to gene transcription, related to tumor cell proliferation and aging [1,2]. Reduced hOGDHc activity is associated with a number of neurodegenerative diseases; however, the link between reductions in mitochondrial TCA cycle enzymes and neurodegeneration has not been established [3,4,5,6]. Among recently reported findings is the involvement of OGDHc in post-translational modifications (PTM) of histones in the nucleus by succinylation/glycosylation in mammalian cells, suggesting a tight link between the metabolic function of OGDHc and the epigenetic regulation of gene expression by PTMs [7,8].

The human 2-oxoadipate dehydrogenase (hE1a, also known as hDHTKD1) is the first component of the 2-oxoadipate dehydrogenase complex (OADHc) in the L-lysine degradation pathway that converts 2-oxoadipic acid (OA) into glutaryl-CoA and NADH (+H^+^) according to the suggested mechanism for OGDHc (Figure 1). In general, members of the 2-oxo acid dehydrogenase complex family have unique E1 and E2 components but share the E3 component. However, the *DLGT* gene encoding the dihydrolipoamide glutaryl-transferase component (hE2a) of hOADHc has not been identified. The Jordan group in vitro [9] and the Houten group in vivo (11) found evidence for the formation of a respective hybrid 2-oxo acid dehydrogenase complex, containing components hE1o, hE2o, hE3, and hE1a, suggesting a potential crosstalk between hOGDHc in the TCA cycle and hOADHc in L-lysine catabolism [9,10]. The notion that two E1s could share a common E2o is certainly unprecedented in the superfamily of 2-oxo acid dehydrogenase complexes and raises the fundamental question: Do hE1a and hE1o interact with the same *loci* on hE2o or different ones? The above findings also raised questions about physiological relevance in health and disease, especially in light of recent genetic findings implicating *DHTKD1* and *OGDHL* (encoding an E1o-like protein) in the genetic etiology of several metabolic disorders [11,12,13,14,15,16,17], such as eosinophilic esophagitis (EoE), a chronic allergic disorder (19), and selected neurodegenerative diseases [18]. At present, there is no available atomic-level structure of any of the 2-oxo acid dehydrogenase complexes. The crystal structure of human E1a has recently been solved with a resolution of 1.9 Å [19] and 2.1 Å [10], both in binary complex with the ThDP cofactor. Recently, the cryo-electron microscopy (cryo-EM) structure of hE1o at 2.92 Å resolution has also been reported [20], but the structure deposited under the PDB ID: 7WGR [20] was not available at the time of the current study and is also missing the first 128 residues, which proved to have functions in the interaction with hE2o. Cryo-EM structures of the inner core domain of hE2o at 5.0 Å [19] and 2.9 Å resolution [21] have recently been reported, revealing a 24-mer cubic architecture that serves as a scaffold for assembly with E1 (E1a and E1o) and E3 [19]. However, the N-terminal region of hE2o (residues 1–150), which comprises the lipoyl domain (LDo) and the linker region (peripheral subunit binding domain), was not observed due to the inherent flexibility of the N-terminus [21]. Due to the lack of an atomic-level structure of an entire OGDHc from any source, owing to the intrinsic structural dynamics of the overall complexes, and of the knowledge about the exact distribution of components around the hE2o core, hydrogen/deuterium exchange (HDX-MS) and chemical cross-linking (CL-MS) mass spectrometry analyses were previously carried out in various binary hOGDHc and hOADHc subcomplexes in the Jordan group [22,23]. The major conclusions from those studies are that an initial formation of the uniquely strong hE1o-hE2o and hE1a-hE2o interaction could facilitate assembly with hE3 into the corresponding complexes and that hE1o and hE1a employ different binding *loci* on hE2o for the formation of the hE1a-hE2o and hE1o-hE2o subcomplexes.

In this paper, we discuss in further detail the assembly of hE1a and hE1o to hE2o *by* applying molecular dynamics simulations to the CL-MS data.

## 2. Results and Discussion

### 2.1. Interaction Loci in Human E1o-E2o and E1a-E2o Binary Subcomplexes Identified by Chemical Cross-Linking Mass Spectrometry

#### 2.1.1. Intercomponent Cross-Links Identified by CDI and DSBU in the hE1o-hE2o Subcomplex

The motivation to establish the loci of interaction in the binary E1-E2 subcomplexes by CL-MS was that neither the reported X-ray structures of the core E2 domains from different sources [24,25,26,27] nor the cryo-EM structure of the hE2o component by itself [21] or in the binary subcomplex E1a-E2o [19] provided structural information regarding the lipoyl domain (LDo) and the subunit-binding region of hE2o presumed to interact with the components hE1o and hE3. In this study, two collision-induced dissociation (CID) cleavable cross-linkers were used: 1,1′-carbonyldiimidazole (CDI, spacer length 2.6 Å) [28] and disuccinimidyl dibutyric urea (DSBU or BuUrBu, spacer length of 12.5 Å) [29]. Taking into account the length of the cross-linkers used and of the Lys side chains (2 × 5.5 Å), as well as the additional distance added to account for the dynamics (7.6 Å) [30] of the backbone, it was assumed that Lys residues within a C–Cα distance of up to 31.1 Å (DSBU) and 21.2 Å (CDI) would be preferred for cross-linking. The different distance constraints imposed by the two cross-linkers were key to gathering more detailed information about the hE1o-hE2o and hE1a-hE2o assemblies. Intersubunit cross-links (Table 1) could be identified with both cross-linkers [31,32].

With CDI, multiple cross-links were identified between Lys^82^ from the N-terminal region of hE1o and Lys^150^, Lys^240^, Lys^289^, and Lys^342^ from the hE2o core domain. The residues Lys^30^ and Lys^34^ are both from a unique peptide comprising the ^26^LENPKSVHKSWDIF^39^ stretch of residues in the N-terminal region of hE1o that experienced a significant delay in deuterium uptake retardation on interaction with hE2o according to HDX-MS studies reported earlier [23]. Importantly, Lys^308^ and Lys^361^ from the ThDP- and Mg^2+^-binding region both formed cross-links with Lys^78^ near the lipoyl domain. Furthermore, Lys^728^ and Lys^959^ from the C-terminal region of hE1o formed cross-links with Lys^24^ and Lys^66^ from the lipoyl domain. Regarding the hE2o component, the three domains—the lipoyl domain (residues 1–77), the linker region (residues 78–151), and the core domain (residues 152–386)—were all involved in cross-linking with hE1o (Table 1). The residues in or near LDo (Lys^24^, Lys^78^, and Lys^87^) were identified as ‘hotspot’ residues in terms of cross-linking. Furthermore, Lys^342^ and Lys^371^ from hE2o, which formed cross-links with Lys^82^ in the N-terminal region of hE1o and Lys^308^ from the ThDP- and Mg^2+^-binding region of hE1o, are located near the active site, where the transthiolsuccinylation reaction results in the formation of succinyl-CoA as a product [31,32]. These data suggest, in good agreement with the general OGDHc catalytic mechanism, that LDo indeed serves as a swinging arm to properly channel the reaction intermediates among the active sites in E1o, E2o, and E3.

In summary, the most prominent interaction loci on hE1o reside in the N-terminal region, the ThDP/Mg^2+^-binding site, and the C-terminal region. It is not surprising that the principal interaction loci on hE2o reside in the core domain, considering that this domain carries out the actual catalytic function of hE2o.

#### 2.1.2. Intercomponent Cross-Links Identified by CDI and DSBU in the hE1a-hE2o Subcomplex

Intersubunit/intercomponent cross-links were identified among Lys^37^, Lys^72^, Lys^110^, Lys^143^, Lys^148^, and Lys^155^ in the N-terminal region of hE1a and several Lys residues from the hE2o core domain (Table 1). In the ThDP/Mg^2+^-binding region, Lys^300^ formed multiple cross-links with hE2o, including Lys^98^ from the linker region as well as Lys^286^ and Lys^373^ from the core domain (Table 1). Importantly, a great number of cross-links, identified by both CDI and DSBU, were detected among Lys^628^, Lys^886^, and Lys^916^ from the C-terminal end of hE1a and Lys^24^, Lys^43^, and Lys^66^ from the LDo, suggesting that the C-terminal region of hE1a could be important for interaction with hE2o and channeling of reaction intermediates (Table 1). Another major difference between the interaction locus patterns in the two subcomplexes studied is that Lys^373^ from the hE2o core domain formed multiple cross-links with Lys residues from the N-terminal region, the ThDP/Mg^2+^-binding region, and the C-terminal region of hE1a, whereas this was not observed in the hE1o-hE2o subcomplex. *This provides strong evidence for different binding modes in the two subcomplexes.* The different number of cross-links across hE2o for the two complexes also points to differences in binding modes, with the hE1a-hE2o complex presenting more cross-links *per* amino acid.

The published X-ray structure of hE1a (PDB ID: 6U3J, ref. [35]) and the cryo-EM structure of hE1o (PDB ID: 7WGR ref. [20]), as well as the hE1o structure from the AlphaFold Protein Structure Database (https://alphafold.ebi.ac.uk/search/text/AF-Q02218-F1 (accessed on 12 September 2021)), were used throughout this study. The sequence alignment (see Figure 1) showed that the two E1 proteins have 36.3% sequence identity and a Cα root-mean-square deviation (RMSD) of 3.97 Å. Comparing the cross-links found for hE1a and hE1o (Table 1), we noted that the linked residues of hE1a and hE1o are generally not equivalent. In no case were the cross-linked residues the ones expected on the basis of structural and sequence-based alignments. The finding that the N-terminal region peptides 26–39 (hE1o) and 24–47 (hE1a), which both displayed decreased deuterium uptake upon binding to hE2o [31,32], were shown to be unique in the two proteins (Figure 1B), also suggested different binding modes for hE1a and hE1o to hE2o.

### 2.2. Structural Modeling of the hE1o-hE2o and hE1a-hE2o Interactions in Binary Subcomplexes by Protein–Protein Docking

For a more in-depth understanding of the interaction networks present, hierarchical clustering using Ward linkage was applied to the docked hE1o-hE2o and hE1a-hE2o structures. In this study, the initial structural models for hE1o and hE2o originated from the AlphaFold Protein Structure Database (IDs: AF-Q02218-F1 (https://alphafold.ebi.ac.uk/search/text/AF-Q02218-F1 (accessed on 12 September 2021)) and AF-P36957-F1 (https://alphafold.ebi.ac.uk/search/text/AF-P36957-F1 (accessed on 12 September 2021))), whereas the hE1a model was downloaded from the Protein Data Bank (ID: 6U3J, ref. [35]). The structures of the starting models were first assembled into dimers (hE1o, hE1a) or trimers (hE2o), according to literature data [21,35], and then energy-minimized to eliminate steric clashes. The above CL-MS information guided the docking experiments; the number of distance restraints to be satisfied was set to three, as some of the measured cross-links might have reflected the interactions of the 24-meric E2o core with multiple E1 units. Distances were evaluated after the final docked structural model was selected by clustering. Next, clustering was carried out on the docked hE1o-hE2o and hE1a-hE2o structures (Figure 2, top). The variance of three hE2o clusters was explored in each clustering study. It was evident that this provided different results for hE1o and hE1a (Figure 2, bottom). Thus, in the hE1o-hE2o structure, the hE2o subunits could reside on either side of the vertical symmetry plane of the hE1o dimer with minor differences in orientation, whereas in the other structure, none of the three hE2o positions provided mirror images (Figure 2A). Neither of the two docked structures satisfied completely the CL-MS data (Appendix A). For the best fit, in the hE1o-hE2o model, 11 of the 31 possible cross-links fell into the expected range. The best docking poses positioned the hE1o dimer onto opposite sides of the symmetrical hE2o trimer (Figure 2A). For the hE1a-hE2o docking model, 20 of the 37 possible cross-links fell within the expected range. In this case, only one cluster contained good candidates for the best structure, as the spacing of the structures was nonsymmetrical. The best orientations for hE1a-hE2o binding showed an identical side/location for the hE2o trimer, with the only difference being the rotation of the hE1a dimer around its vertical symmetry axis (Figure 2B).

The structural models of the hE1o-hE2o and hE1a-hE2o subcomplexes revealed some similarities in their interactions. The core domain always binds to E1 with the lipoyllysyl arm positioned near the ThDP/Mg^2+^-binding regions. The structural models showed that interactions are mostly localized in the hE2o core region encompassing residues 229–244 (when interacting with hE1o) or 269–289 (with hE1a) based on previous HDX-MS [23,32] and current MD simulation results. The N-terminal regions, comprising hE1a residues 24–47 and hE1o residues 26–39, are protected from solvent accessibility by the hE2o core, rather than being directly involved in binding to hE2o, according to findings reported in the literature [19]. In hE1a, the linker region 485–518 is involved in binding to hE2o, but the corresponding hE1o region (residues 525–558, according to structural alignment, rather than residues 530–558 in pairwise sequence alignment) is not located in proximity to the interaction site on hE2o.

The important conclusion from these studies is that the best orientations of hE1o and hE1a for their interactions with hE2o are markedly different: for the hE1o-hE2o subcomplex, a model is suggested in which the hE1o components are placed on opposite sides of the symmetric hE2o trimer, with the N-terminus and the α/β1 domain providing the principal interactors. In the hE1a-hE2o subcomplex, an identical localization of the hE2o trimer and a rotation of the hE1a dimer around its vertical symmetry axis are proposed, with the interdomain linker region and the α/β1 catalytic domain interacting with the trimer. Nevertheless, both the hE1o and hE2o possess very flexible N-termini, and this flexibility introduces a certain degree of uncertainty to the conclusions.

### 2.3. Molecular Dynamics Simulations

For the complexes studied, we observed cross-links of the LDo domain of hE2o with both the N-termini and C-termini of the E1 proteins. To satisfy the observed interactions, the internal motions of the LDo and the linker domain of hE2o must also be considered. This was not feasible in the course of the docking. Therefore, next, molecular dynamics simulations were performed in an attempt to gain further insight into the interactions. The starting structures for the simulations were uncertain, as the Alphafold models were used. This is most observable for hE2o, where residues 77–153 have a low to very low confidence; LDo has a high, while the remainder of the structure has a very high confidence. For hE1o, the Alphafold structure is of very low confidence only in the N-terminal region (residues 27–51). Therefore, our goal with the simulation was not to obtain an atomic-level, spatially (coordinate-wise) absolutely accurate description of the respective complexes, but rather a coarse interpretation of similarities and differences between the two subcomplexes. Because of the uncertainties in the starting structures and since literature data indicated that loop-opening motions are in the range of 10– 20 ns [38], we have selected 50 ns simulations. Overall, the simulation results are in accord with our previous hydrogen/deuterium exchange and cross-linking mass spectrometry data.

#### 2.3.1. Analysis of the Trajectories Calculated for the N-Terminal Region of hE2o

MD simulations enabled us to analyze the molecular motions of the N-terminal region of hE2o and identify its temporal interactions with hE1. In particular, the study addressed the flexibility around the N-terminal Lys^43^ in LDo, which is covalently modified by lipoic acid and forms a lipoyl-lysine arm essential for active-site coupling and substrate channeling. By assuming that the functional dynamics of the LDo affect the flexibility of the essential Lys^43^, we analyzed the RMSD index for the displacement of the Lys^43^ Cα during a simulation of 50 ns. MD simulations for each binary complex hE1 dimer–hE2o trimer showed that these systems practically stabilized after 5 ns, with minor fluctuation of the protein backbone. Analysis of the molecular motions throughout the trajectory revealed that the Lys^43^ Cα underwent a ping-pong-like motion with a high degree of displacement (32.53 Å average RMSD and 2.26 Å RMSF (root-mean-square fluctuation)). RMSD is a measure of the difference between two structures, where the reference structure is the starting point in the simulation and the target structure is the structure obtained at any molecular dynamics step of interest. In MD simulations, one is interested in how a structure and its parts change over time as compared to the starting point. RMSD is used to identify rather large changes in a protein structure as compared to the starting point. The leveling or flattening of the RMSD curve may indicate that the protein structure has stabilized and equilibrated. RMSF is also a numerical measure; but instead of indicating positional differences between entire structures over time, RMSF is a measure of the individual residue flexibility, or how much a particular residue moves (fluctuates) during a simulation.

The large RMSD and RMSF values obtained for the lipoyl and linker domains highlight their extensive motions, which allow the lipoic acid carrying Lys^43^ to move roughly as much as the dimension of a single core domain (Figure 3A). The displacement was more pronounced for the hE1a-hE2o complex (average RMSD 76.94 Å), but for only one chain of the hE2o trimer. The fluctuation of the Lys^43^ position, as seen in the trajectory, could also play a role in the catalytic function of the protein. To discover whether it does, we evaluated the trajectory using normal mode analysis (NMA), which is a frequently applied technique in protein dynamics studies to probe the flexible states accessible to a protein near the equilibrium position [19,39]. NMA can be used, for example, to predict functional motions, i.e., motions that relate to protein function. In this study, we assumed that the motions of the N-terminal region in hE2o, including the LDo and the linker region, are the ones that are of the greatest relevance to the function of hE2o; in particular, the channeling of the reaction intermediates between the active sites of hE1o and hE2o. The NMA applied to the entire hE2o revealed a multitude of shape changes (flexible states) accessible to the N-terminal region in hE2o, the extent of which was analyzed in detail. Several normal modes that belong to an unstructured N-terminal region could be identified. The conformational motion with the highest magnitude was attributed to a hinge-bending motion of the peptide encompassing residues 78–133, which also moves the LDo (and Lys^43^) when not bound. Figure 3B reveals the trajectory of motion for peptide 78–133. In the extended conformation of the region encompassing residues 78–133, the interaction with hE1 is feasible, whereas the other extreme conformation might serve for its interaction with hE3 (not analyzed). This motion should initiate interactions with other proteins, because this zone must be relocated before the lipoic acid can interact with a different partner. Overall, the MD analysis suggested that the conformational states of the N-terminal region (residues 78–133, in particular) in hE2o are potentially capable of promoting interactions with hE1 and hE3.

#### 2.3.2. The MD Simulation of the hE1a Dimer–hE2o Trimer Subcomplex

A different way to analyze MD results is to visualize the motions along the trajectory (see the LDo motion video in Appendix B, trajectory of hE1a-hE2o, same as Figure 4). Simulations were carried out using trimeric hE2o, and it is apparent that the molecular motions of the three monomers are not superimposable. The case of the hE1a-hE2o simulation is discussed below. Similar dynamics were observed for the hE1o-hE2o complex as well.

Although the hE2o trimer has three lipoyl domains (LDo), they are not all needed for the interaction with an E1 dimer. During the simulation, three cases were observed: an E1 bound to hE2o chain, a ‘free’ hE2o chain that interacts only with other hE2o chains, and an hE2o chain, where an LDo temporally interacts with hE1a. The translational motions of an unbound LDo are shown in Figure 4, where hE1a and hE2o were modeled. The starting position of the LDo is close to the hE2o trimer. From there, a nonlinear motion is seen, which results in an interaction with the 485–518 region in hE1a (in lilac). A similar motion is expected to lead to the cross-links measured for the C-terminal region in hE1a. When comparing flexibility in hE1o and hE1a (Appendix A), a lower flexibility of the interdomain linker region and a floppier C-terminus could be observed in hE1a, relative to hE1o. The motion of the C-terminal hE1a peptide (899–919), which is not a conserved feature in all 2-oxo acid dehydrogenases but is present in most of the mammalian OGDHc E1 components, is also apparent during the MD simulation, showing a temporary interaction and spatial proximity with the LDo of hE2o. The hE1a C-terminus acts as a guide for the LDo motion and maintains interaction with it, as seen in the 45 ns snapshot of the simulation (Figure 4). The key peptides identified for the dynamics of hE1a-hE2o have their own counterparts in hE1o, with the hE1a stretch 485–518 corresponding to the shorter hE1o stretch 530–558.

The motions of the hE2o LDo correlate with the conformational changes of both E1 proteins studied. To better describe the conformational changes of the complexes, we studied the fate of the native contacts for the two complexes. Native contacts exist within the native state of the protein (here the starting state), as opposed to nonnative contacts, which are formed during conformation changes. Calculating the fraction of native contacts (Q) over a simulation shows whether transitions between states or unfolding processes are observed. For both simulations, a shift between two states could be observed. The Q1 and Q2 plots (Appendix A) show the fractions of the initial contacts that survived the transition; for both hE1a and hE1o, this number is approximately 80%. Changes in folding states can also be identified and visualized by principal component analysis. Figure 5 shows the projection of two main components colored according to the simulation time. Analyzing the hE2o trimer (Figure 5A), three distinct states could be observed which belong to the conformational transitions of the different chains of the protein. The other component of the complex, the two E1 chains, also have asymmetrical dynamics (Figure 5B).

#### 2.3.3. Comparison between hE1a and hE1o

The HDX-MS data previously reported by us demonstrated that the N-terminal region in hE1o, specifically the two peptides ^18^YVEEM^22^ and ^26^ENPKSVHKSWDIF^39^, experienced a significant deuterium uptake retardation when interacting with hE2o [23]. The MD simulations point to the same characteristics, particularly that the hE1o N-terminal peptide 18–39 is shielded through the hE1o-hE2o interaction. Analysis of the H-bonds in the course of the simulation pointed to the hE2o 96–153 region as a stabilizing feature of the complex, as it participates in forming the most H-bonds with various regions of the E1 proteins. In hE1o, several residues form temporary H-bonds, but the highest population is attributed to the conserved residues 58–61. For hE1a, the primary interaction is not at the N-terminus; Arg^246^, Pro^898^, Phe^179^, and Gln^506^ proved to participate in the most stable H-bonds. Several other H-bonds in the interdomain linker region (480–521) were also detected, albeit with a lower population. These findings explain the cross-links detected at the same time for distant parts of the E1a protein, with the same E2o residue. The contact maps of both hE1(o or a)-hE2o complexes, calculated in the course of the MD simulations, pointed out that the hE2o linker region is of equal importance for both E1 proteins, but the core region 229–244 interacts with hE1o 143–145 and 200–207, while the hE2o residues 269–289 interact with hE1a 271–277. In addition, the number of contacts increased significantly for hE1a, resulting in a lower flexibility of the α/β2 domain (Appendix A).

To gain more insight into the interactions of hE1o-hE2o and hE1a-hE2o, the residue-level dynamics was evaluated as averaged root-mean-square fluctuations (RMSF) for the backbone atoms in hE1o and hE1a. This revealed greater flexibility (higher RMSF values) for the respective N-terminal regions, the interdomain linker regions that connect the respective two halves of the hE1 subunits [19,20], and for residues in the corresponding α/β2 domains involved in binding the ThDP cofactors (Appendix A). The RMSF plots indicated that, on interaction with hE1o or hE1a, the hE2o linker region exhibited higher RMSF values compared to other regions of hE2o (Figure 6). This high relative flexibility also shows the importance of the hE2o linker region for the stability and function of the implicated enzyme complexes: The maintained flexibility ensures the possibility of interaction with all the respective components of the overall complexes.

To obtain a quantitative representation of the interfaces between the respective E1 and E2 proteins, overall binding ΔG values were calculated for the complexes before and after the simulations. The starting ΔG values were −15.0 kcal/mol for the hE1o-hE2o complex and −10.9 kcal/mol for the hE1a-hE2o complex. Binding ΔG values were also calculated on representative structures from the ends of the simulations. For both complexes, stabilization of the starting structure was observed. The ΔG values increased to −38.4 kcal/mol for hE1o-hE2o and −33.4 kcal/mol for hE1a-hE2o. These values are in line with the observed stabilities of the complexes.

The following could be concluded here: (1) As was observed in previous HDX-MS studies and current MD simulations for both complexes, the N-terminal peptides in the hE1 proteins (18–39 in hE1o, 24–47 in hE1a) are shielded from the solvent, but do not interact directly with hE2o. (2) The hE2o linker region exhibits the highest number of H-bonds with the E1 components; but in the case of hE1o, the N-terminus and α/β1 helix are mainly involved in the interaction, while for hE1a the interdomain linker and α/β1 helix have the most stable contacts during the simulation. This conclusion is in line with the observed cross-links for the two complexes. (3) Although the two E1 proteins have rather similar structures, E1-E2 interactions were not observed at conserved or equivalent residues. (4) The C-termini are involved in the dynamic interactions of the proteins studied. At least two conformations of the complexes were present in solution, as were indicated by the detected cross-links and dynamics study.

## 3. Materials and Methods

### 3.1. Reagents

ThDP, NAD^+^, CoA, DTT, isopropyl 1-thio-β-D-galactopyranoside, imidazole, thiamin-HCl, glycerol, DNase I, and Micrococcal nuclease were from Affymetrix (Thermo Fisher Scientific, Waltham, MA, USA); 2-oxoglutaric acid, 2-oxoadipic acid, iodoacetamide, benzamidine HCl, disuccinimidyl dibutyric urea (DSBU or BuUrBu), 1.1′-carbonyl-diimidazole (CDI), and dimethyl sulfoxide were from Sigma-Aldrich (St. Louis, MO, USA). Formic acid solution was obtained from Fluka Analytical (Buchs, Switzerland); MS grade trypsin protease was from Pierce; Ni Sepharose^®^ high-performance affinity media was from GE Healthcare (Chicago, IL, USA).

### 3.2. Protein Expression and Purification

The expression and purification of the human components E1o, E1a, and E2o were as reported earlier [9,40,41].

### 3.3. Chemical Cross-Linking Mass Spectrometry

#### 3.3.1. Sample Preparation for Chemical Cross-Linking with CDI and DSBU

Two cross-linkers, DSBU and CDI, were employed for cross-linking of E1o or E1a with the E2o component, as follows. First, hE1o or hE1a (1 nmol, 33 µM concentration of subunits) and hE2o (1 nmol, 33 µM concentration of subunit) were mixed in 15 µL of 20 mM K_2_HPO_4_ (pH 7.5) containing 0.5 mM ThDP, 1 mM MgCl_2_, 0.15 M NaCl, and 10% glycerol. After 30 min of incubation at 20 °C, 1 µL of CDI (1 M) dissolved in DMSO was added and the cross-linking reaction was conducted at 15 °C for 45 min. When using DSBU as a cross-linking reagent, 1 µL of DSBU (150 mM) dissolved in DMSO was added and the reaction was conducted at 37 °C for 20 min. To quench the reaction, the reaction mixture was diluted to 50 µL with the reaction buffer above, and 1 M Tris-HCl (pH 8.0) was added to a final concentration of 20 mM Tris-HCl. Cross-linking efficiency was assessed by SDS-PAGE (7.5%).

#### 3.3.2. Tryptic and Glu-C Proteolysis

Cross-linked samples were subjected to tryptic and Glu-C in-solution double digestion. An aliquot containing 1 nmol of total protein was withdrawn from each reaction mixture and placed into 70 µL of 8 M urea in 100 mM NH_4_HCO_3_, and the samples were incubated at 60 °C. After 20 min of incubation, 2 µL of 200 mM DTT was added and the samples were incubated for an additional 40 min. Next, 3.5 µL of 200 mM iodoacetamide was added and the samples were incubated at room temperature in the dark for 30 min. Then 1 µL of 200 mM DTT was added, with a subsequent dilution of the reaction mixture with 0.85 mL of 100 mM NH_4_HCO_3_. Glu-C digestion was carried out in a 50:1 protein to Glu-C (wt/wt) ratio at 37 °C. After 4 h of incubation, the reaction mixtures were subjected to tryptic digestion in a 100:1 protein to trypsin ratio at 37 °C. After overnight digestion, the reaction was terminated by adding 2 µL of 95% formic acid. The digested samples were desalted on a SepPak SPE C-18 column (Waters), dried using a SpeedVac centrifuge (Thermo Scientific Savant, Waltham, MA, USA), and then dissolved in 100 µL of 20% acetonitrile (0.05% formic acid).

#### 3.3.3. Enrichment of Cross-Linked Products

Peptide enrichment was performed using an SCX trap cartridge (Optimize Technologies, Inc., Oregon City, OR, USA). The digested samples were desalted on a SepPak SPE C-18 column, dried in a SpeedVac, and then reconstituted in 100 µL of 20% acetonitrile/0.05% formic acid. The enrichment was carried out using an SCX trap cartridge (Optimize Technologies, Inc.). The trap was washed with 5 volumes of 20% acetonitrile, 0.4 M NaCl, and 0.05% formic acid and equilibrated with 5 volumes of 20% acetonitrile and 0.05% formic acid. Then, after loading the samples three times in the trap, the peptides were released from the trap by using 20% acetonitrile/0.05% formic acid with increasing concentrations of NaCl: 4 volumes of 10 mM, 2 volumes of 40 mM, 4 volumes of 200 mM, and 4 volumes of 400 mM NaCl. Fractions eluted with 200 and 400 mM NaCl, providing most of the cross-linked products, were collected, concentrated in vacuo, and then redissolved in H_2_O containing 0.1% formic acid.

#### 3.3.4. Analysis of the Cross-Linked Peptides by Nano-LC-MS/MS

The enriched peptides were desalted using a SepPak SPE C-18 column and dried using a SpeedVac centrifuge. Cross-linked peptides were analyzed by nano-LC-MS/MS (Dionex Ultimate 3000 RLSC nano system interfaced with Q Exactive HF (Thermo Fisher Scientific, Waltham, MA, USA)). The samples were loaded into a self-packed 100-μm by 2-cm trap (Magic C18AQ; 5 μm and 200 Å; Michrom Bioresources, Inc., Auburn, CA, USA) and washed with buffer A (0.1% trifluoroacetic acid) for 5 min at a flow rate of 10 μL/min. The trap was brought in line with the analytical column (self-packed Magic C18AQ; 3 μm and 200 Å; 75 μm by 50 cm), and peptides were eluted at 300 nl/min using a segmented linear gradient of 4%–15% solution A (0.2% formic acid) for 30 min, followed by a 15%–25% gradient of solution B (0.16% formic acid and 80% acetonitrile) for 40 min and continued with a 25%–50% solution B for 44 min and 50%–90% of solution B for 11 min. Mass spectrometric data were acquired using a data-dependent acquisition procedure with a cyclic series of a full scan with a resolution of 120,000, followed by MS/MS (higher-energy C-trap dissociation; relative collision energy: 27%) of the 20 most intense ions and a dynamic exclusion duration of 20 s.

#### 3.3.5. Analysis of Cross-Linking MS Data

LC-MS/MS peak lists were generated using the ProteoWizard 3.0.18156 [41] software package and searched against the SwissProt database (proteome ID: UP000000558, https://www.ebi.ac.uk/interpro/proteome/uniprot/UP000000558/ (accessed on 3 May 2020)). The search was performed using search GUI-3.3.117 incorporated with Peptideshaker (1.16.42) [42,43]. The search parameters were as follows: fragment mass error, 20 ppm; parent mass error, 5 ppm; fixed modification, carbamidomethylation on cysteine; potential modifications during the initial search, methionine oxidation and acetylation on protein N-terminus; and up to one missed tryptic cleavage. This search showed that no contaminant proteins above 5% were present in the samples. Cross-linked products were evaluated using the Merox 1.6.6 software tool ([34] with the following parameters: precursor, 5 ppm; fragment, 15 ppm mass accuracy; CDI was set as a cross-linker). Only Lys-Lys cross-linked peptides were analyzed. The cross-linked peptides were manually validated, and the results with the False Discovery Rate (FDR) < 0.01 limit were analyzed (FDR was calculated by Merox software (version 1.6.6) [34]). Data were visualized using the xiView website [44].

### 3.4. Structural Modeling by Docking the hE1o-hE2o and hE1a-hE2o Structures

Docking was performed using the protocol incorporated in the Schrödinger suite program Glide version 8.8 [45]. Briefly, a series of hierarchical filters were used to find potential locations for a ligand around a receptor. A series of conformations were examined during docking, and OPLS2005 (GLIDE HTVS) with a distance-dependent dielectric model was used to refine the most promising orientations. A small number of orientations, which collected the best Emodel scores, were reported as a final list. The length of the list, as an input parameter, was determined by us, the users.

Input preparation for docking included restrained minimization, refinement of the starting structure (for hE1a with missing residues), and protonation state assignments using the PROPKA algorithm [46]. After these steps, potential disulfide (for cysteines closer than 3.2 Å) and H-bond assignments were carried out.

After docking, the results were clustered in the Schrodinger suite using Ward clustering. The Ward method is a hierarchical clustering method; thus, no preconception is applied to the data, and it is model-free [47]. The resulting list of the docking structures was cut up to 30 structures, which may represent well the dynamic nature of the hE1o-hE2o and hE1a-hE2o subcomplexes. Another selection criterion was that the cross-linked distances, together with potential clashes, were evaluated on the 24-meric cubic structure of hE2o. Considering that collisions with the 24-meric hE2o core domain should be minimal, approximately half of the structures could be eliminated. The remaining structures were evaluated on the basis of the Cα–Cα distances obtained for cross-linked Lys residues by CL-MS.

### 3.5. Molecular Dynamics Simulation Analysis

#### 3.5.1. Input Preparation

A quantitative measure of flexibility and the magnitudes of molecular motions in the *N*-terminal region were obtained by 50 ns of molecular dynamics simulation. The molecular dynamics simulation used as a starting point one of the cluster centers from the docking step. Molecular graphics software Visual Molecular Dynamics (VMD) [48] and UCSF Chimera [49] were used to generate coordinate and protein structure (.psf) files applying the atom types and connectivity definitions of the CHARMM36m force field [50]. The system was set up in a way that the ionization states of the amino acid side chains mimicked the pH~7 condition, and the δ-N of the histidine imidazole ring was protonated (HSD residue). This was set up by the PROPKA algorithm [46] already in the docking workflow. The starting structure for hE2o was not the published 6H05 [21] structure alone, due to the missing LDo, but the trimeric hE2o core of the 6H05 structure and the respective Alphafold structure superimposed, which showed an overall RMSD of 1.195 Å.

#### 3.5.2. MD Simulation and Analysis

A parallelizable molecular dynamics package, the Nanoscale Molecular Dynamics (NAMD) program (Beckman Institute for Advanced Science and Technology at the University of Illinois at Urbana-Champaign) [51], was used to run the simulations of these complexes. Simulations were carried out under Linux using the CUDA-enabled version 2.14 of the NAMD software package. The solvent and ionic strength of the medium were treated using the generalized Born Implicit Solvent model implemented in NAMD (GB-OBC1 model). In terms of time steps, 1 fs intervals were applied. The parameters were set up for the simulations as follows: electrostatic interaction, a cut-off distance of 12 Å, and a grid width of 1 Å. The switching distance for non-bonded electrostatics and van der Waals interactions was 10 Å, and an exclusion policy “scaled 1-4” was used. The final temperature of the system was set to 310.15 K. The conformational space sampled by the protein is dependent on temperature. As such, we have tried to replicate physiological conditions. The energy minimizations of the constructed protein complexes were first performed using the conjugate gradient method for 250,000 cycles. Equilibrations were run for microcanonical (NVE) ensembles with the temperature increasing from 10.15 K to the final temperature by 10 K every 2000-simulation step for a total of 250,000 cycles. The quality of equilibration was monitored by calculating the energy distribution of the ensembles and fitting it to a Boltzmann distribution; the temperature distribution was also calculated. The MD simulation was performed in the isobaric–isothermal (NPT) ensemble. The temperature was maintained at 310.15 K, while the pressure was constant at 101,325 Pa (1 atm). Langevin Dynamics was used, which enables pressure control via the Langevin piston Nose–Hoover algorithm [52]. Analysis of the resulting trajectories and normal modes was carried out in VMD with the help of the ProDy plugin (version 2.0) [53]. Anisotropic network modeling together with the principal component analysis [54,55] was used to define harmonic modes of oscillation and their magnitudes. Principal component analysis was performed in Python version 3.10 separately on the E1 and E2 proteins along the simulation trajectory. The output of the analysis was limited to three components, and the results were visualized as orthogonal projections of the principal components and colored according to the simulation time. The stability of the simulation was monitored by calculating the RMSD of the correctly folded core domains of hE2o (see Appendix A for the RMSD plot of the hE2o core domain). Overall binding ΔG values for the interfaces of the respective E1 and E2 proteins were calculated using the PDBePISA server [56].

## 4. Conclusions

The hE1o-hE2o interaction by CL-MS. Analysis of the hE1o-hE2o subcomplex by CL-MS revealed the most prominent *loci* for their interaction. In hE1o, Lys^30^, Lys^34^, and Lys^82^ from the N-terminal region were identified to interact with hE2o Lys^85^ and Lys^87^ from the LDo and Lys^150^, Lys^240^, Lys^289^, and Lys^342^ from the hE2o linker region and the core domain. These findings are consistent with earlier HDX-MS studies showing that two N-terminal regions in hE2o, ^18^YVEEM^22^ and ^26^LENPKSVHKSWDIF^39^, experienced significant retardation in deuterium uptake when interacting with the full-length hE1o [23].

The hE1a-hE2o interaction by CL-MS. CL-MS identified a number of cross-links in the C-terminal region of hE1a, which is different from those identified in hE1o, suggesting that the C-terminal region could be important for the interaction with hE2o. Another major difference is that Lys^286^ and Lys^373^ from the hE2o core domain formed multiple cross-links with Lys residues from the N-terminal region, the ThDP- and Mg^2+^-binding region, and the C-terminal region in hE1a, not seen in the hE1o-hE2o subcomplex, suggesting a different binding mode.

Modeling and dynamics of the binary hE1o-hE2o and hE1a-hE2o subcomplexes based on distances identified by CL-MS and MD simulations led to the following conclusions: (1) Clustering also predicted different binding modes for hE1o and hE1a: the hE1o dimers were placed at opposite sides of the symmetric hE2o trimer, whereas the hE1a dimers, when bound, rotated around their vertical symmetry axes. The N-terminal regions in hE1a (residues 24–47) and hE1o (residues 26–39) are protected from solvent accessibility, but are not directly bound by the hE2o core. In addition, these stretches of amino acids are neither sequentially nor structurally equivalent to each other according to sequence and structural alignments (see Figure 1). (2) The hE2o linker region is of great importance for the stability and dynamic operation of the relevant complexes. At the same time, the hE2o core is in spatial proximity to the hE1 α/β1 domains; the MD simulation showed that the hE2o residues 229–244 are important for the hE1o interaction, while residues 269–289 are seen in the contact map of hE1a-hE2o. The hE1o residues 143–145 and 200–207 and hE1a residues 271–277 interact with the hE2o core region. (3) The hE1-hE2 interactions are present in the hE1 α/β1 domains, but the two hE1 proteins are not equivalent, no conserved residue pairs were observed to interact throughout the simulation. (4) Considerable dynamics were observed in the C-termini for both hE1o and hE1a, and these provide further interaction surfaces for both of the hE1-hE2o complexes. Protein dynamics also showed the availability of multiple conformations for each complex.

## Data Availability

The mass spectrometry proteomics data have been deposited to the ProteomeXchange Consortium (www.proteomexchange.org (accessed on 28 February 2020)) via the PRIDE partner repository with the dataset identifiers PXD017792 and PXD023525.

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
