# Peer review of "Probing the E1o-E2o and E1a-E2o Interactions in Binary Subcomplexes of the Human 2-Oxoglutarate Dehydrogenase and 2-Oxoadipate Dehydrogenase Complexes by Chemical Cross-Linking Mass Spectrometry and Molecular Dynamics Simulation"

_ijms, 2023, doi:10.3390/ijms24054555_

Round 1

Reviewer 1 Report

in my opinion the work is well done and merits publication

Reviewer 2 Report

When I reviewed the manuscript, I realized that it was not strictly my field of expertise.

Since I am rather a synthetic organic chemist with peptide chem and modelling background, I could not confidently rank or judge the novelty from the biological aspect. Although, the topic presented, convinced me about the appropriate scientific level.

After a detailed reading, I accept the manuscript in the present form, after some minor improvements of the figures and I suggest it for publication.

This manuscript presents the combination of a few experimental and theoretical methods. The applied docking and MD simulations seems to be correct, however, there was no validation of the theoretical method or I did not find a citation to validate it. I missed specification od the version of the Glide program used (or year of release).

 The Figures of the manuscript represent adequate quality, except of Figures 3 and 4. They are transparent and clear with these exception. Figure 3, represent the result of the MD is under the level of the ms. For Figure 4, it was disturbing to me, that a small part of the protein is cut off from the picture (e.g. 3. 5. 6.). It degrades somewhat the high quality.

The same is true also for the quality of the table 1 as well, it could be improved. Moreover, its story could be summarized in a more concise way.

 Overall, the manuscript is well written and contains no typographical errors. The text is fluid and readable, although lots of abbreviations are included due to the topic.

The number of 51 References seems to be correct and enough to described method. I found 7 self-citations out of 51, but on closer inspection they are valid citations.

Reviewer 3 Report

The human 2-oxoglutarate dehydrogenase complex, a key enzyme of TCA cycle, comprise three components: hE1o, hE2o, and hE3. In general, members of 2-oxo acid dehydrogenase have unique E1/E2 but share E3. The E1 component of 2-oxoadipate dehydrogenase (hE1a) has been identified, whereas the E2 component of 2-oxoadipate dehydrogenase has not identified. Interestingly, hybrid 2-oxo acid dehydrogenase complex composed of hE1o, hE2o, hE3, and hE1a was identified.

To characterize the binding modes of hE1o/hE2o/hE1a, authors tried to evaluate the interaction between hE1o and hE1a/hE1o by cross-linking mass spectrometry (CL-MS) and molecular dynamics simulation. By the CL-MS methods, authors identified several cross-linking between hE1o and hE1a/hE1o. Together with the previous HDX-MS results, authors concluded that N-terminal flexible region of hE1o is important to interact with hE2o core/catalytic domain and LDo domain, whereas C-terminal region of hE1o is important to interact with hE2o. Thus the binding mode of hE1a/hE2o would be different from hE1o/hE2o although the overall structure of hE1a is similar with hE1o.

Authors further carried docking simulation of hE1a/hE1o dimer and hE2o trimer incorporating CL-MS result as a restraining parameter. The result also suggests that the binding mode of hE1a/hE2o would be different from that of hE1o/hE2o although the docking structures did not completely satisfy the CL-MS results.

Authors further carried out molecular dynamics simulations of the putative hE1a/hE2o and hE1o/hE2o complexes. Simulation clearly suggests that the LDo domain is highly mobile due to the flexibility of the linker region. Thus LDo domain can transport lipoyl acid from E1 core to E2 core.

The story is attractive, but the simulation is based on the alphafold structure of hE2o. The model confidence of the flexible linker region is reported to be low in the database. Therefore, the simulation is not so accurate. At least, the model confidence of the alphafold structure of the LDo domain and the linker region of hE2o must be noted in the text although the alphafold structure of the core domain is resemble to the cryo-EM structure.

Otherwise, the results appear to be interesting, but the manuscript contains several concerns as shown below. The manuscript is incomplete.

1.    The results of the CL-MS data are summarized in Table 1. The authors reported the interaction between E1 and E2 based on the table, but it is difficult to associate with the structural features of E1 and E2. I recommend color-coding by domain in Table1; e.g. the N-terminal region of E1 is blue, the lypoil domain of E2 is red, the core domain of E2 is green. I have also noticed that the residue numbers in the sections 2.1.1 and 2.1.2 differ from those in Table 1. The cross-link between Lys 83 of hE1o and Lys342 of hE2o is reported in the text (line 40, page 3), but Lys82 of hE1o is described in Table 1. The crosslink between Lys37 of hE1a and hE2o is reported in the text (line 5, page 4), but Lys37 of hE1a is not found in Table 1. The lipoyl domain is defined between residues 1-77 in line 37, page 3, but Lys78 and Lys87 are included in the lipoyl domain in line 38. Please correct them.

2.    Sequence alignment and superimposition of the structures of hE1o and hE1a are shown in Figure 1. Please indicate which is the N-terminal and C-terminal side of the structure in panel A, and show the disordered parts of the crystal structure of hE1a and the cryo-EM structure of hE1o in panel B.

3.    Please explain the color-coding and difference between model coupling and model docking in Figure 2. To easily compare panel A and panel B, please align the orientation of the E2o core. In the current figure, it is difficult to see that they are bound to different sites. It is better to use the same color-coding of E2o in panels A and B.

4.    The distances in Fig. S1 are given to a three decimal places, but the accuracy of the atomic coordination is not so high even if it was a crystal structure.

5.    Please explain the object shown in the appendix. Is this a trajectory of hE1a-hE2a or hE1a-hE2o? If it is same as shown in Figure 4, why is the color-coding different?

6.    In section 4.2, the authors noted the preparation of E3, but it is not found in the results and discussion. Is this used for expression and purification of other components?

7.    In page 8, lines 13, the sentence describes about hE1o-hE2o complex. Please correct Fig. 2B to Fig. 2A.

Reviewer 4 Report

The authors adopt the chemical cross-linking mass spectrometry and molecular dynamics simulations to probe assembly of hE1a and hE1o in binary subcomplexes. The results provide positive and interesting information. This work can be accepted in international journal of molecular science after minor revision.

1. Why is the temperature of the simulation systems set as 310.15 K? The authors should simply clarify the reason.

2. Why is the δ-N of the histidine imidazole ring protonated? The authors should simply explain the reason.

3. The details of molecular docking should be richened.

4. The details on how to perform principal component analysis should be added, moreover two works on principal component analysis (J. Chem. Inf. Model. 2021, 61, 1954−1969 and J. Chem. Inf. Model. 2022, 62, 6118−6132) should be mentioned.

5. The authors performed 50-ns molecular dynamics simulations, in fact this simulation time is short for a big system. Thus the authors should simply discuss the reliability of their simulations.

Reviewer 5 Report

Attached file

Round 2

Reviewer 3 Report

Authors incorporated all my suggestions in the revised manuscript. I recommend to publish the manuscript in the present form.

Author Response

The authors thank  Reviewer 3 for the careful and useful comments to help make our manuscript easier to read.

Reviewer 5 Report

Attached file.
